# Research on Physicochemical Properties and In Vitro Digestive Characteristics of High-Amylose Corn Starch–Ultrasound-Treated Waxy Rice Flour Blends

**DOI:** 10.3390/foods14162920

**Published:** 2025-08-21

**Authors:** Yuxing Wang, Yu Guo, Zhiting Zhu, Yan Ding, Yuchan Yang, Dongxu Wang, Zhanming Li, Yuanxin Guo, Xiaoman Chen

**Affiliations:** 1Jiangsu Provincial Engineering Research Center of Grain Bioprocessing, School of Grain Science and Technology, Jiangsu University of Science and Technology, Zhenjiang 212100, China; wyuxing2023@163.com (Y.W.); gy18234591259@163.com (Y.G.); zzting2025@163.com (Z.Z.); dingyan010306@163.com (Y.D.); 221211803113@stu.just.edu.cn (Y.Y.); wdx@just.edu.cn (D.W.);; 2College of Basic Medical Sciences, Bengbu Medical University, Bengbu 233030, China

**Keywords:** ultrasound-treated waxy rice flour, high-amylose corn starch, blends, physicochemical properties, in vitro digestion, rice cake

## Abstract

This study aimed to investigate the effect of high-amylose corn starch (HACS) addition on the physicochemical properties and in vitro digestibility of an ultrasound-treated waxy rice flour (UWRF)–HACS blend system. As the proportion of HACS increased, the amylose content in the blends significantly increased (*p* < 0.05), while their water solubility index (WSI) and swelling power (SP) significantly decreased (*p* < 0.05). Additionally, the average particle size of the blends increased, and the surface of starch granules became smoother. Compared to UWRF, the blends did not generate new functional groups, but increased the starch’s relative crystallinity and short-range ordered structure. Rheological results indicated that the HACS-UWRF blends were mainly elastic and exhibited a typical weak gel system. In vitro digestibility results showed that the addition of HACS significantly increased the resistant starch (RS) content in the rice cakes (*p* < 0.05), while substantially reducing the hydrolysis index (HI) and estimated glycemic index (eGI) (*p* < 0.05). This study revealed the processing characteristics and gelatinization behavior changes in the HACS-UWRF blends. It provides a theoretical basis for the development of specialized flour for slow-glycemic rice cakes.

## 1. Introduction

Glutinous rice (*Oryza sativa* L.), also known as sticky rice, is renowned for its high stickiness, strong viscosity, and excellent edibility. Its products are highly popular among consumers in Southeast Asia [1]. However, glutinous rice starch contains up to 98% amylopectin, which leads to a rapid increase in blood sugar after consumption of glutinous rice products. Thus, it is not suitable for people with high blood sugar or related sub-health groups [2]. With the popularization of the concept of healthy living, how to modify glutinous rice starch to reduce its glycemic index (GI) has gradually become a hot topic in high-value product development.

Most current research focuses on improving the functional properties of starch through physical, chemical or biological means. However, due to issues such as residual chemical reagents, complex processes and high costs [3], the application of chemical modification and enzymatic modification methods in the food-processing field has been limited to a certain extent. In contrast, physical modification techniques do not require exogenous chemical substances, are easy to operate and have high safety; therefore, they better meet the green development needs of the food industry. Among these, ultrasonic treatment has become a research focus in recent years for its high efficiency and controllability [4]. Yang et al. [5] found that after ultrasonic treatment, the surface of glutinous rice starch granules became concave. The content of amylose significantly increased from 0.51% to 1.91% (*p* < 0.05), and the content of resistant starch (RS) significantly increased from 4.03% to 7.69% (*p* < 0.05). Florese et al. [6] demonstrated that ultrasonic treatment of glutinous rice starch could alter its digestive properties and increase the amylose content. Our research group also conducted a modification study on glutinous rice flour by using ultrasonic treatment. The results showed that under specific conditions (30% emulsion concentration, 400 W ultrasonic power, 4 min ultrasonic time and 49 °C drying temperature), ultrasonic treatment can promote transformation among components of glutinous rice starch and increase RS content. However, the estimated glycemic index (eGI) value of the ultrasound-treated waxy rice flour (UWRF) we obtained was 76.31. According to the classification of glycemic index (low GI ≤ 55, medium GI 56–69, high GI ≥ 70) [7], UWRF is still classified as a high-GI food [8]. How can the eGI of UWRF be further reduced? We hypothesize that, based on ultrasonic modification, a simpler and more efficient method may be to further add high-amylose compounds.

The amylose content in high-amylose corn starch (HACS) is greater than 70%. Compared with common starch, HACS exhibits stronger heat resistance. After high-temperature treatment, more RS components are retained, conferring higher anti-digestive properties and a lower GI [9]. In recent years, HACS has been widely used in food preparation due to these advantages. Ma et al. [10] found that noodles made by adding different proportions of HACS to wheat flour had higher tensile strength and extensibility. Yuan et al. [11] observed that the gelling and thickening abilities of dough kneaded by blending HACS and wheat starch in a 1:1 ratio were improved, and the RS content was significantly increased (*p* < 0.05). Gu et al. [12] found that when HACS was used to partially replace wheat flour to prepare steamed buns, the RS content significantly increased from 12.01% to 28.49% (*p* < 0.05), while the eGI value significantly decreased from 94.39 to 57.55 (*p* < 0.05). Studies have confirmed that amylose content is significantly positively correlated with RS and significantly negatively correlated with eGI (*p* < 0.05) [13]. Although there are studies on adding HACS to food, few reports have focused on its application in ultrasound-treated foods. We hypothesized that combining these two physical methods (ultrasound and HACS compounding) would help explore changes in the physicochemical properties and in vitro digestibility of the blends. This work is of great significance for guiding the development of slow-glycemic specialized glutinous rice flour through in-depth research on the variation patterns of the blends.

In this study, we aimed to progressively incorporate HACS into UWRF to produce specialized glutinous rice flour. From the perspective of investigating the powder properties of HACS-UWRF binary blends, we systematically analyzed the variation patterns of physicochemical properties, including particle structure (crystalline structure, short-range ordered structure, and particle morphology, etc.), rheological properties, gelatinization characteristics, thermal properties, and in vitro digestibility. This research provides theoretical backing for the development of novel slow-glycemic specialized glutinous rice flour.

## 2. Materials and Methods

### 2.1. Materials

Glutinous rice flour was provided by Changzhou Jintan Jiangnan Flour Co., Ltd., (Changzhou, China), and Xiangyu 1945 high-amylose corn starch (amylose content ≥ 70%) was purchased from Shanghai Koch Biotechnology Co., Ltd., (Shanghai, China). Potato amylose standard solution and potato amylopectin standard solution were obtained from Shanghai Xibiao Technology Co., Ltd., (Shanghai, China). Sodium hydroxide, methanol, anhydrous ethanol, acetic acid, potassium iodide, iodine solution and sodium dodecylbenzenesulfonate were acquired from Shanghai Guoyao Group Chemical Reagent Co., Ltd., (Shanghai, China). 3,5-dinitrosalicylic acid was procured from Shanghai Macklin Biochemical Technology Co., Ltd., (Shanghai, China). Amyloglucosidase (100,000 U/mL) and thermostable α-amylase (2000 U/mL) were purchased from Aladdin Biochemical Technology Co., Ltd., (Shanghai, China). All chemicals and enzymes used in this study were of analytical grade.

### 2.2. Pre-Treatment of Raw Materials

Referring to a previous study by our group [8], when the concentration of glutinous rice flour emulsion was 30%, the ultrasonic conditions were as follows: frequency 25 kHz, time 4 min, power 400 W, and drying temperature 49 °C. The resulting product was used as UWRF for subsequent experiments. HACS was added to UWRF at ratios of 0%, 5%, 10%, 15%, 20%, 25% and 30% (*w*/*w*, based on the dry weight of UWRF) and thoroughly mixed. These blends were sequentially labeled as 0% H, 5% H, 10% H, 15% H, 20% H, 25% H and 30% H.

### 2.3. Amylose Content of HACS-UWRF Blends

Our research group referred to the Chinese national standard GB/T 15683-2025 “Inspection of grain and oils–Determination of amylose content in rice” [14]. The steps were as follows: the sample was crushed and sieved, then defatted by refluxing with methanol and dried. The sample (0.1 g) was weighed into a centrifuge tube and mixed with ethanol (1 mL) and NaOH (9 mL, 1 mol/L), and the starch was dissolved in a boiling water bath for 10 min. After cooling and centrifugation, the supernatant was transferred to a 100 mL volumetric flask and made up to volume. Then, the diluent (5 mL) was taken from it, followed by the addition of iodine solution (2 mL) and acetic acid (1 mL) and the mixture was made up to volume again in a 100 mL volumetric flask. After thorough shaking, the mixture was allowed to react in the dark for 10 min. Absorbance was measured at 720 nm (based on the specific color reaction of the complex formed by amylose and iodine). The amylose content of the sample was calculated using the amylose standard curve (y = 0.0054 x + 0.0887, R^2^ = 0.9928).

### 2.4. Water Solubility Index and Swelling Potential of HACS-UWRF Blends

Referring to the method described by Derycke et al. [15], a sample (0.5 g) was weighed into centrifuge tubes, mixed with distilled water (25 mL) and shaken for 1 h. The mixture was then stirred at 160 r/min for 30 min using a thermostatic magnetic stirrer (DF-101T, Bio-Tek, Winooski, VT, USA) at 95 °C. After cooling to room temperature, the mixture was centrifuged at 4000 r/min for 15 min. The supernatant was poured into an aluminum box and dried in a 105 °C oven until it reached a constant weight. The wet precipitate in the centrifuge tube was weighed. The equations used for calculating the water solubility index (WSI) and swelling power (SP) of the samples are as follows:$$\mathrm{Water}\;\mathrm{solubility}\;\mathrm{index},\;\mathrm{WSI}\;(\%)=W_{2}{/W}_{1}\times 100$$$$\mathrm{Swelling}\;\mathrm{power},\;\mathrm{SP}(\%)=W_{3}{/W}_{1}(100-\mathrm{WSI})\times 100$$
where W_1_ is the mass of rice flour, g; W_2_ is the dry weight of the supernatant; W_3_ is the mass of the wet precipitate, g.

### 2.5. DSC of HACS-UWRF Blends

Referring to the method of Kong [16], a sample (3 mg) was weighed into an aluminum crucible, and deionized water was added at a ratio of 1:2 (*w*/*v*). The crucible was compacted and sealed, and then equilibrated at 4 °C in a refrigerator. Thermal properties of the samples were determined using a differential scanning calorimeter (DSC 25, TA Instruments-Waters, New Castle, DE, USA) under the following conditions: temperature range 25–130 °C, heating rate 10 °C/min and nitrogen flow rate 50 mL/min. Data were analyzed using STAR Evaluation Software (v15.01, Mettler Toledo, Greifensee, Switzerland).

### 2.6. RVA of HACS-UWRF Blends

An appropriate amount of HACS-UWRF blend powder was placed in a sample can, and a specific proportion of deionized water was added. The mixture was stirred thoroughly to ensure complete immersion of the sample in water, after which the aluminum can was placed in a Rapid Visco Analyzer (RVA 4800, Perten Instruments, Stockholm, Sweden) for analysis. The RVA program was set as follows: equilibration at 50 °C for 1 min, then heated at a rate of 6 °C/min to the target temperature of 95 °C and maintained for 5 min, then cooled at the same rate to 50 °C and held for 2 min. The gelatinization curve of the composite powder was then recorded.

### 2.7. Rheological Properties of HACS-UWRF Blends

#### 2.7.1. Determination of Dynamic Viscoelastic Properties

Frequency scan: referring to the method of Gong et al. [17]. The sample was mixed with an appropriate amount of water to prepare a 6% emulsion by mass fraction. After 30 min in a boiling water bath, the determination was carried out using a rotational rheometer (Discovery HR10, TA Instruments, Waters, DE, USA). A certain amount of emulsion was taken and quickly transferred onto a preheated ø 40 mm plate, and sealed with silicone oil. The spacing was set at 1 mm and the strain at 0.5% (in the linear viscoelastic zone). Test procedure: after the emulsion was equilibrated at 25 °C for 3 min, frequency was scanned from 0.1 to 100 Hz.

#### 2.7.2. Determination of Static Rheological Properties

The sample emulsion was prepared and placed on a preheated ø40 mm plate, sealed with silicone oil, with an interval of 1 mm and a set temperature of 25 °C. Steady-state shear tests were conducted with the shear rate increasing by 0.1 to 100 s^−1^ and decreasing by 100 to 0.1 s^−1^. The apparent viscosity values and the variation curves of the shear rate were recorded.

### 2.8. FTIR of HACS-UWRF Blends

The method was slightly modified based on Yang [18], the samples were mixed with potassium bromide (KBr) at a mass ratio of 1:100, ground and pressed into thin slices and then put into a Fourier transform infrared spectrometer (FTIR, American Thermoelectric Co., Ltd., Chicago, IL, USA) for full-band scanning (400 to 4000 cm^−1^ ) with three scans and a resolution of 4 cm^−1^. Baseline correction, smoothing and deconvolution of the infrared spectra were carried out using Omnic and PeakFit Version 4.12 software to obtain the R1047 cm^−1^/1022 cm^−1^ and R1022 cm^−1^/995 cm^−1^ ratios of the samples.

### 2.9. X-Ray Diffraction (XRD) Patterns of HACS-UWRF Blends

The sample was sieved through a 0.075 mm sieve and placed in the XRD (SmartLab, Rigaku Corporation, Akishima-shi, Japan) for scanning. The testing conditions are as follows: target type: Cu radiation, voltage: 40 kV, current: 150 mA, scanning range: 5° to 40°, step size: 0.02°, scanning speed: 6°/min and measurement of the 2θ X-ray diffraction spectrum. Use MDI Jade 6 software to calculate the sample’s relative crystallinity.$$\mathrm{Relative}\;\mathrm{crystallinity},\;\mathrm{RC}\;(\%)=\frac{\mathrm{Ac}}{\mathrm{Ac}+\mathrm{Aa}}\times 100$$
where RC is the relative crystallinity of the sample, %; Ac is the area of the crystalline zone of the sample and Aa is the area of the amorphous zone of the sample.

### 2.10. SEM Analysis of HACS-UWRF Blends

A small amount of conductive adhesive was glued on the sample stage, and the sample was vacuum-sputter coated with gold. The sample was then observed using a field-emission scanning electron microscope (SEM, Zeiss Merlin Compact, Carl Zeiss, Oberkochen, Germany). The sample was observed and photographed at an accelerating potential of 10 kV.

### 2.11. Particle Size Distribution of HACS-UWRF Blends

Referring to the method of Deng [19] with a slight modification, mix the sample with deionized water to prepare a 1% emulsion. Then, an appropriate amount of the emulsion sample was added to the sampler of the laser particle-size analyzer (Dandong Baxter Instrument Co., Ltd., Dandong, China), and ultrasonically oscillated for 2 min to evenly distribute the starch particles and achieve a shading degree of 10% to 15%. The stirring speed was set to 3000 rpm and the refractive indices of water and the sample to 1.33 and 1.52, respectively.

### 2.12. In Vitro Digestion of HACS-UWRF Blends

We referred to the methods of Englyst [20] and Wang et al. [2] for the in vitro simulated digestibility test. Fresh rice cakes were refrigerated for 12 h, after which 200 mg of sample was mixed with 15 mL of 0.2 mol/L sodium acetate buffer (pH 5.2) and incubated in a water bath for 30 min. A 10 mL mixture of amyloglucosidase (100,000 U/mL) and thermostable α-amylase (2000 U/mL) was then added, and enzymatic hydrolysis was conducted in a 37 °C water bath. At hydrolysis times of 0, 10, 20, 40, 60, 90, 120 and 180 min, 1 mL aliquots of the hydrolyzate were transferred to test tubes. 4 mL of ethanol was added to inactivate the enzyme and the tubes were centrifuged at 10,000 r/min and 4 °C for 5 min. The precipitate was discarded, and the supernatant was diluted 10-fold.

The glucose content in glutinous rice flour samples was determined by the DNS method. DNS reagent (2 mL) was added to the dilution solution (1 mL), and the mixture was heated in a boiling water bath for 5 min. After cooling and further dilution, the absorbance was measured at 520 nm. The glucose content was calculated through the glucose standard curve (y = 0.9765 x + 0.0383, R^2^ = 0.9992), and the hydrolysis curve of the glutinous rice flour sample was plotted. The area under the curve reflected the effect of in vitro digestion of the food on blood glucose. Taking white bread (with the hydrolysis rate of white bread defined as 100) as the control standard, the content of nutrient fragments, rapidly digestible starch (RDS), slowly digestible starch (SDS), RS, hydrolysis rate, hydrolysis index (HI) and eGI values were calculated from the following formulas:$$\mathrm{RDS}(\%)=\frac{{(G}_{20}-\mathrm{FG})\times 0.9}{\mathrm{TS}}$$$$\mathrm{SDS}(\%)=\frac{{(G}_{120}-G_{20})\times 0.9}{\mathrm{TS}}$$$$\mathrm{RS}\;(\%)=\frac{\mathrm{TS}-(\mathrm{RDS}+\mathrm{SDS})}{\mathrm{TS}}$$$$\mathrm{Hydrolysis}\;\mathrm{rate}\;(\%)=\frac{\mathrm{Gt}\times 0.9}{200}$$$$\mathrm{Hydrolysis}\;\mathrm{index},\;\mathrm{HI}=\frac{\mathrm{Area}\;\mathrm{under}\;\mathrm{the}\;\mathrm{sample}\;\mathrm{digestion}\;\mathrm{curve}}{\mathrm{Area}\;\mathrm{under}\;\mathrm{the}\;\mathrm{digestion}\;\mathrm{curve}\;\mathrm{of}\;\mathrm{reference}\;\mathrm{standard}\;\mathrm{samples}}\times 100$$eGI=39.71+0.549H
where G_20_ and G_120_ indicate the glucose content at 20 min and 120 min, respectively. FG represents the free glucose content, and TS is the total starch content; G_t_ indicates the glucose content during 0 to 180 min.

### 2.13. Statistical Analysis

All experiments were performed in triplicate. Statistical analysis was performed using IBM SPSS 19.0, including one-way analysis of variance (ANOVA) and Duncan’s multiple range test. Differences were considered significant at *p* < 0.05, and results were expressed as mean ± standard deviation. Graphs were generated using OriginPro 2021.

## 3. Results and Discussion

### 3.1. Effect of HACS Addition on the Amylose Content, Water Solubility Index and Swelling Power in HACS-UWRF Blends

The amylose content directly affects the crystal structure, thermal properties and gelatinization characteristics of the blends [21], while the water solubility index (WSI) and swelling power (SP) can indirectly reflect the internal forces and water-holding capacity of starch particles [22]. As shown in Table 1, with the addition of HACS, the amylose content in the blends significantly increased from 3.82% to 16.78% (*p* < 0.05). This is conducive to reducing the digestion rate of starch and enhancing the stability of postprandial blood glucose [23]. Previous studies have confirmed that amylose content is negatively correlated with the WSI and SP [24]. The results showed that as the amylose content increased, the WSI significantly decreased from 22.90% to 6.17% (*p* < 0.05), while the SP also decreased significantly from 19.63% to 14.02% (*p* < 0.05). This phenomenon can be attributed to the entanglement between amylose and amylopectin, which forms a stable crystalline network structure. This structure not only reduces the leaching of soluble substances, but also effectively inhibits starch granule swelling [25]. Similar findings were reported by He et al. [26] in their study on HACS addition to potato starch.

### 3.2. Effect of HACS Addition on Thermodynamic Properties of HACS-UWRF Blends

DSC mainly reflects the energy change in the blends and starch thermal stability during the melting process of starch crystals [27]. As shown in Figure 1 and Table 2, the UWRF onset temperature (T_o_) was 62.23 °C, peak temperature (T_p_) was 86.55 °C, and conclusion temperature (T_c_) was 95.43 °C. With the increase in HACS addition, the peaks shifted to higher temperatures and broadened in shape. The T_o_, T_p_ and T_c_ all increased significantly (*p* < 0.05). Many studies have noted that T_o_, T_p_ and T_c_ are correlated with amylose content and molecular order during starch granule phase transition [28,29]. This suggests that a higher amylose content strengthens hydrogen bonding between starch granules and other components, requiring higher temperatures for gelatinization. Gelatinization enthalpy (ΔH) represents the energy needed for starch gelatinization. A higher ΔH indicates a more ordered starch molecular structure and greater compactness of starch granule architecture [30]. With the increase in HACS addition, ΔH increased significantly from 14.82 J/g to 18.34 J/g (*p* < 0.05), indicating that enhanced amylose interactions in the blends improved starch crystalline structure stability, necessitating higher temperatures to disrupt starch granule crystalline regions [31]. This result is consistent with the findings of Wang et al. [32].

### 3.3. Effect of HACS Addition on Gelatinization Properties of HACS-UWRF Blends

Gelatinization properties are one of the sensitive indicators of starch-based raw materials, determining the cooking quality and edible quality of food. They are closely related to the content of amylose content, starch chain length distribution, and starch crystal structure [33]. As shown in Figure 2 and Table 3, with an increase in HACS, the gelatinization temperature of starch rises, while peak viscosity (PV), trough viscosity (TV) and final viscosity (FV) all decrease significantly (*p* < 0.05). PV significantly decreases from 3386.00 mPa·s to 2317.50 mPa·s (*p* < 0.05), which aligns with the findings of Yu et al. [34]. The addition of HACS enhances the rigidity of starch granules, facilitating the formation of a spatial barrier effect during gelatinization and inhibiting the water absorption and expansion capacity of starch granules [35], which is consistent with the hydration property results. Breakdown (BD) is a critical indicator of starch paste thermal stability and shear resistance. Lower BD values indicate stronger thermal stability and shear resistance of the starch paste [36]. With an increase in HACS, BD significantly decreased from 1604.50 mPa·s to 815.50 mPa·s (*p* < 0.05), indicating that the introduction of HACS can alleviate the rupture of starch granules in the blends and enhance thermal stability, which is mutually consistent with the changing trends of starch PV and TV. These different gelatinization characteristics influence diverse applications of starch in the food industry. This low-viscosity and thermally stable starch property has better adaptability in the production and processing of products such as cakes and bread [37]. The setback (SB) reflects the cold paste stability of starch. With the addition of HACS, SB significantly decreased from 606.00 mPa·s to 395.00 mPa·s (*p* < 0.05), indicating that the blends are less prone to retrogradation and hardening during temperature reduction, and more likely to form a gel with higher strength and better elasticity after cooling. Therefore, it exhibits enhanced edible quality and an extended shelf life [38].

### 3.4. Effect of HACS Addition on the Rheological Properties of HACS-UWRF

#### 3.4.1. Effect of HACS Addition on the Dynamic Viscoelastic Properties of HACS-UWRF Blends

Frequency scanning is one of the most commonly used methods for studying the rheological properties of starch [39]. As shown in Figure 3, within the linear viscoelastic range, the storage modulus (G′), loss modulus (G″) and loss angle tangent (tanδ) of each blends increase with angular frequency, and G′ > G″, tanδ < 1. This indicates that the starch paste is mainly elastic with weakened viscosity, belonging to a typical elastic solid weak gel system [40]. Studies have shown that the increase in G′ can be attributed to the increase in amylose content, indicating that the addition of HACS enhances the cross-linking and polymerization degree of amylose and amylopectin molecules, making the internal structure of starch more stable [41]. This is consistent with the research results of Moreira et al. [42].

A further comparison of G′, G″, and tanδ values for each sample at a frequency of 40 Hz is provided in Table 4. Within the same frequency scanning range, G′ significantly increased from 17.28 Pa to 23.00 Pa (*p* < 0.05), and G″ significantly increased from 8.71 Pa to 11.19 Pa (*p* < 0.05). The presence of amylose restricts the expansion and gelatinization of starch granules, enhances the entanglement and interaction of starch molecular chains and improves the anti-deformation ability of starch paste [43], which is consistent with the results of gelatinization characteristics.

#### 3.4.2. Effect of HACS Addition on the Static Rheological Properties of HACS-UWRF Blends

The apparent viscosity of starch paste refers to the ratio of shear stress to shear rate under a certain velocity gradient, and its change affects the textural properties of starch-based foods [44]. As shown in Figure 4, when the shear rate is 0 s^−1^, with the addition of HACS, the initial viscosity of the blend system decreases. As the shear rate increases, the apparent viscosity of the blends shows an obvious downward trend, and UWRF-HACS exhibits shear thinning behavior, which is a typical characteristic of pseudoplastic non-Newtonian fluids [45]. A similar phenomenon is also observed in the study on the structure of japonica glutinous yellow rice starch [46]. This phenomenon indicates that when the mixed powder is in a static state or under a low shear rate, the amylose and amylopectin molecular chains intertwine with each other, forming high viscosity resistance, thereby hindering the flow of the starch molecules; while under a high shear rate, the spatial network structure of the starch paste is disrupted, resulting in a decrease in the flow resistance between molecules, and subsequently leading to a reduction in viscosity [47].

### 3.5. Effect of HACS Addition on the Short-Range Ordered Structure of HACS-UWRF Blends

FTIR is mainly used to study the surface structure of starch granules. The absorption peaks near 3440 cm^−1^ reflect the O-H stretching vibration peaks in starch, and those near 2930 cm^−1^ represent the C-H stretching vibration [48]. After the addition of HACS, the infrared spectrum of the blends was similar to that of UWRF, with no new absorption peaks appearing (Figure 5). This suggests that blends with different masses of HACS did not generate new functional groups or alter the chemical structure of the starch, consistent with the research results of Zhao et al. [49]. However, changes in peak intensity and width indicate that the HACS addition affected hydrogen bonds between starch molecular chains and altered the short-range ordered structure [48].

As shown in Table 5, with the increase in HACS, the ratio of R1047 cm^−1^/1022 cm^−1^ significantly increased from 0.69 to 0.97 (*p* < 0.05), indicating that HACS helps form a more stable ordered structure, with an increase in the content of double helix structures and a tighter internal structure of starch granules [50]. R1022 cm^−1^/995 cm^−1^ significantly decreased from 2.60 to 1.50 (*p* < 0.05). The decreased amylopectin content reduced the degree of interlacing between starch molecular chains and the structure of the amorphous region [51].

### 3.6. Effect of HACS Addition on the Crystalline Structure of HACS-UWRF Blends

Starch, as a natural polycrystalline system, is mainly composed of crystalline regions and amorphous regions, which correspond, respectively, to the sharp peaks and diffuse peaks in X-ray diffraction patterns [52]. Blends with different HACS additions all exhibit typical A-type crystalline characteristics [53]. Prominent diffraction peaks are observed at 2θ angles of 15°, 17°, 18° and 23°, while small diffraction peaks can be seen at 9°, 11°, 27°, 31° and 34° (Figure 6a). Relative crystallinity (RC) reflects the long-range ordered structure of starch during retrogradation [54]. As shown in Figure 6b and Appendix A, with increasing HACS addition, the RC of the blends significantly increases from 16.30% to 24.54% (*p* < 0.05), which is consistent with the observation by Zhang et al. [55] that HACS increases the RC of buckwheat noodles. Studies have shown that amylose is more likely to form starch–lipid complexes than amylopectin [56]. Higher amylose content results in a higher degree of starch polymerization and a more stable starch structure [57], which aligns with the trends of R1047 cm^−1^/1022 cm^−1^ and R1022 cm^−1^/995 cm^−1^ in FTIR analysis.

### 3.7. Effect of HACS Addition on the Microstructure of HACS-UWRF Blends

The microscopic morphology of blends with different HACS addition amounts was assessed. UWRF particles exhibit an irregular polyhedral shape with varying sizes. The surface of the starch particles has small holes and depressions, and the structure is relatively loose. This might be due to mechanical damage to the starch particles caused by ultrasonic treatment, consistent with the research results of Fashi et al. [58]. With increasing HACS content, the macromolecular substances in the system are observed to increase continuously, and the surface of starch particles also becomes smoother (Figure 7). Natural HACS particles are irregular in shape (e.g., spherical and polygonal), small and uniform. After addition, they promote the generation of more amylose, which repolymerizes and rearranges, making the internal structure of starch molecules more compact and the surface pores smaller [50]. This conclusion is consistent with the FTIR analysis results.

### 3.8. Effect of HACS Addition on Particle Size Distribution of HACS-UWRF Blends

Particle size is one of the critical indicators of powder products. Research shows that the size of starch particles is related to the physicochemical properties of starch-based raw materials, such as amylose content, expansion potential, thermal stability, and gelatinization properties [59]. As HACS addition increases, the particle sizes of all samples exhibit a multi-peak distribution, specifically four particle size peaks (Figure 8). Since the particle size of HACS is generally larger than that of UWRF, the average particle size of the blends gradually increases from 118.13 μm to 124.20 μm (Table 6), consistent with the phenomenon observed by SEM. This increase in particle size and change in aggregation state reduce the contact area between particles and water, inhibiting the permeation of water molecules and the dissolution of starch chains [60], ultimately resulting in a significant decrease in WSI and SP (*p* < 0.05).

### 3.9. Effect of HACS Addition on the In Vitro Digestive Characteristics of Rice Cake

Rice cakes were prepared by adding an appropriate amount of water and edible oil to the HACS-UWRF blends. The effect of HACS addition on the in vitro digestibility of rice cakes is shown in Figure 9a,b. With increasing HACS content, the RDS content of rice cakes significantly decreased from 23.77% ± 0.46 to 13.16% ± 0.63 (*p* < 0.05), while the RS content significantly increased from 53.57% ± 0.65 to 63.83% ± 0.53 (*p* < 0.05), and the SDS content showed no significant change (Appendix A). The increase in RS inhibits the activities of α-amylase and amyloglucosidase, effectively stabilizing postprandial blood glucose fluctuations [61]. It indicates that HACS and UWRF combine closely, forming a relatively regular crystalline structure between starch molecules. The starch crystalline regions create a physical barrier to the biological accessibility of digestive enzymes [62], consistent with the XRD results. The HI significantly decreased from 54.64 ± 0.28 to 35.06 ± 0.03 (*p* < 0.05), and the eGI value significantly decreased from 69.16 ± 0.15 to 58.97 ± 0.01 (*p* < 0.05) (Appendix A), indicating that the rice cake is a medium-GI food (55 < GI < 70) [7]. Previous studies have confirmed that HI and eGI values obtained via in vitro simulation methods are highly correlated with the actual in vivo GI in humans. However, in vitro models cannot fully replicate in vivo dynamic processes (e.g., hormone regulation, the role of the gut microbiota), so eGI results require further verification through animal experiments or human trials [63].

## 4. Conclusions

The addition of HACS significantly affected the physicochemical properties and in vitro digestibility of the HACS-UWRF blend system. With increasing HACS content, amylose content significantly increased (*p* < 0.05), which compacted the internal structure of starch molecules, gradually increased the average particle size, and smoothed the surface of starch particles. Meanwhile, HACS alleviated the rupture of starch particles in the blends, inhibiting starch particle water absorption and expansion. This reduced the gelatinization viscosity of the mixed powder, ultimately enhancing the thermal paste stability and shear resistance of the starch and suppressing starch retrogradation. XRD and FTIR results further indicated that all blends exhibited typical A-type crystallization. HACS addition enhanced the RC and short-range ordered structure of starch. Rheological results showed that the starch paste of the blends is a pseudoplastic non-Newtonian fluid, exhibiting shear-thinning behavior with the system being mainly elastic. In vitro digestibility results revealed that with increasing HACS content, the RS content of rice cakes significantly increased from 53.57% ± 0.65 to 63.83% ± 0.53 (*p* < 0.05), and the eGI value significantly decreased from 69.16 ± 0.15 to 58.97 ± 0.01 (*p* < 0.05). Rice cakes belong to the category of medium-GI foods. These findings can be applied to develop glutinous rice starch foods with slow-glycemic increase properties.

## Figures and Tables

**Figure 1 foods-14-02920-f001:**
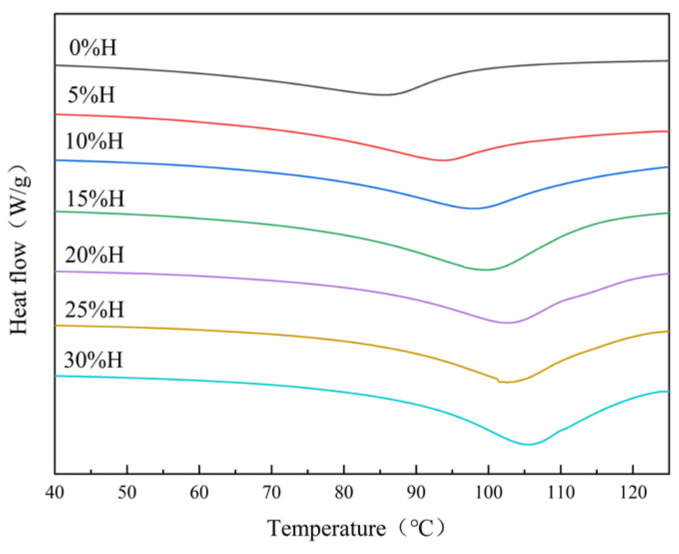
Effect of different levels of HACS addition on the differential thermogram of the HACS-UWRF blends.

**Figure 2 foods-14-02920-f002:**
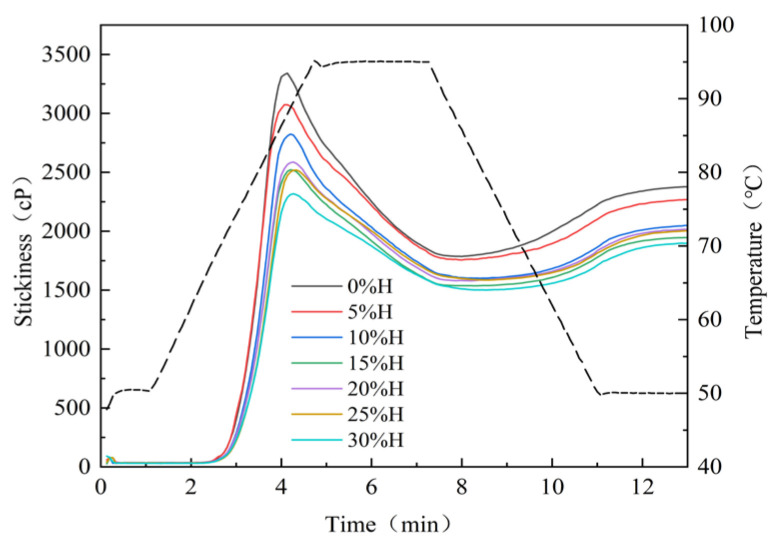
Effect of different levels of HACS addition on the gelatinization curve of the HACS-UWRF blends.

**Figure 3 foods-14-02920-f003:**
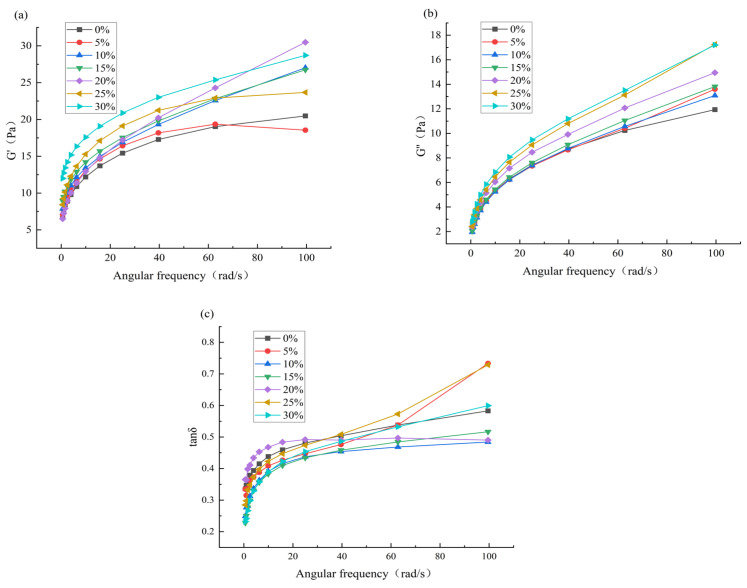
Effect of different levels of HACS addition on G′ value (**a**), G″ value (**b**) and tanδ (**c**) in the frequency scan of the HACS-UWRF blends, respectively.

**Figure 4 foods-14-02920-f004:**
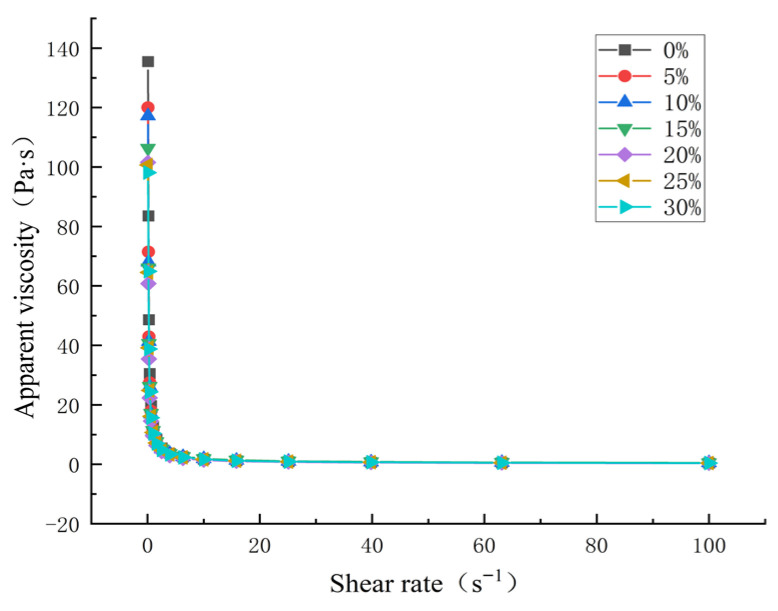
Effect of HACS addition on apparent viscosity of HACS-UWRF blends.

**Figure 5 foods-14-02920-f005:**
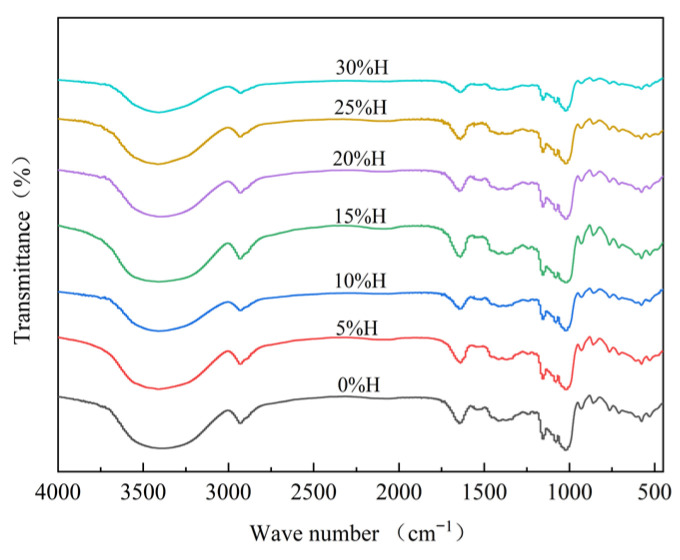
Effect of different levels of HACS addition on the short-range ordered structure of HACS-UWRF blends.

**Figure 6 foods-14-02920-f006:**
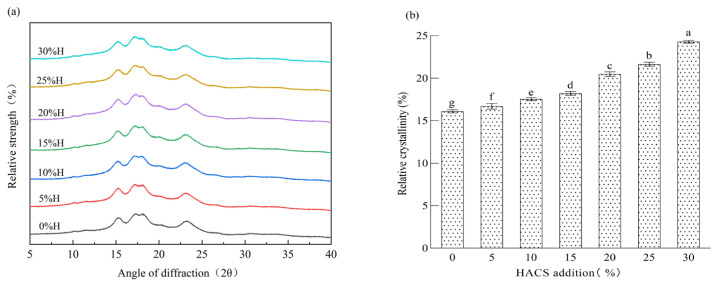
Effect of different levels of HACS addition on X-ray diffraction patterns of HACS-UWRF blends (**a**) and changes in relative crystallinity content (**b**). Values with different letters in the graph are significantly different (*p* < 0.05).

**Figure 7 foods-14-02920-f007:**
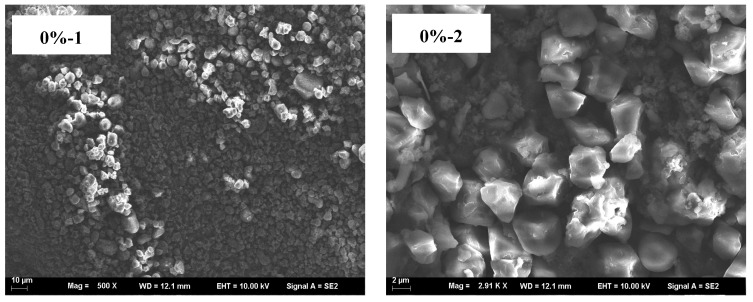
SEM micrograph of the samples; (1) magnification of the left column is 500×, and (2) magnification of the right column is 3000×.

**Figure 8 foods-14-02920-f008:**
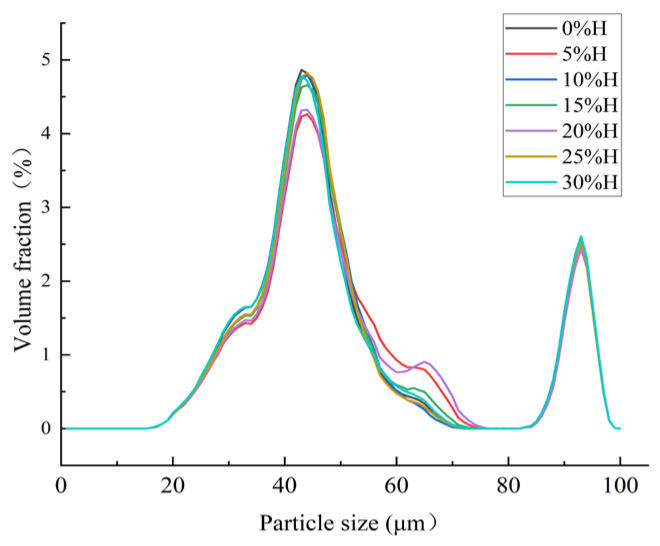
Particle size distribution of different levels of HACS addition to HACS-UWRF blends.

**Figure 9 foods-14-02920-f009:**
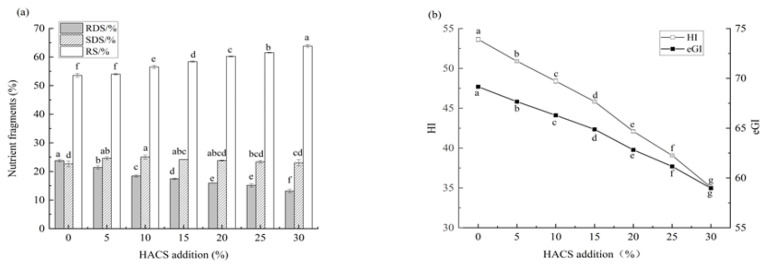
Effect of different levels of HACS addition on RDS, SDS and RS contents (**a**) and eGI and HI values (**b**) of rice cake. Values with different letters in the graph are significantly different (*p* < 0.05).

**Table 1 foods-14-02920-t001:** Effect of different levels of HACS addition on the content of amylose, water solubility index and swelling power in HACS-UWRF blends.

HACS Addition/%	Amylose Content (%)	WSI (%)	SP (%)
0	3.82 ± 0.30 ^g^	22.90 ± 0.75 ^a^	19.63 ± 0.54 ^a^
5	7.09 ± 0.12 ^f^	18.43 ± 0.49 ^b^	18.35 ± 1.77 ^ab^
10	9.47 ± 0.28 ^e^	14.72 ± 0.97 ^c^	17.49 ± 0.79 ^bc^
15	11.75 ± 0.52 ^d^	12.00 ± 0.01 ^d^	16.22 ± 0.72 ^cd^
20	12.67 ± 0.21 ^c^	10.37 ± 0.15 ^e^	16.03 ± 0.38 ^cd^
25	14.99 ± 0.30 ^b^	8.70 ± 0.26 ^f^	15.12 ± 0.22 ^de^
30	16.78 ± 0.87 ^a^	6.17 ± 0.15 ^g^	14.02 ± 0.38 ^e^

Values with a different letter in the same column are significantly different (*p* < 0.05).

**Table 2 foods-14-02920-t002:** Effect of different levels of HACS addition on thermodynamic parameters of HACS-UWRF blends.

HACS Addition/%	T_o_ (°C)	T_p_ (°C)	T_c_ (°C)	ΔH (J/g)
0	62.23 ± 0.12 ^g^	86.55 ± 0.37 ^g^	95.43 ± 0.30 ^f^	14.82 ± 0.52 ^g^
5	74.37 ± 0.18 ^f^	93.54 ± 0.21 ^f^	105.52 ± 0.35 ^e^	16.23 ± 2.55 ^f^
10	76.35 ± 0.27 ^e^	98.29 ± 0.26 ^e^	106.66 ± 0.20 ^d^	16.91 ± 0.49 ^e^
15	78.30 ± 0.24 ^d^	100.33 ± 0.46 ^d^	109.35 ± 0.36 ^c^	17.26 ± 0.40 ^d^
20	84.60 ± 0.33 ^c^	102.15 ± 0.90 ^c^	112.56 ± 0.31 ^b^	17.55 ± 2.46 ^c^
25	87.46 ± 0.22 ^b^	103.39 ± 0.48 ^b^	114.44 ± 0.32 ^a^	17.71 ± 0.77 ^b^
30	92.74 ± 0.11 ^a^	105.23 ± 0.38 ^a^	114.79 ± 0.22 ^a^	18.34 ± 1.00 ^a^

Values with a different letter in the same column are significantly different (*p* < 0.05). T_o_, T_p_, and T_c_ indicate the temperature of the onset, peak, and conclusion of gelatinization, respectively.

**Table 3 foods-14-02920-t003:** Effect of different levels of HACS addition on gelatinization characteristics of HACS-UWRF blends.

HACS Addition/%	PV/ (mPa·s)	TV/ (mPa·s)	FV/ (mPa·s)	BD/ (mPa·s)	SB/ (mPa·s)
0	3386.00 ± 54.00 ^a^	1781.50 ± 35.50 ^a^	2387.50 ± 37.50 ^a^	1604.50 ± 18.50 ^a^	606.00 ± 2.00 ^a^
5	3090.00 ± 3.00 ^b^	1756.50 ± 7.50 ^a^	2268.00 ± 7.00 ^b^	1333.50 ± 10.50 ^b^	511.50 ± 0.50 ^b^
10	2825.00 ± 10.00 ^c^	1599.50 ± 2.50 ^c^	2048.50 ± 2.50 ^d^	1225.50 ± 12.50 ^c^	449.00 ± 5.00 ^cd^
15	2761.00 ± 8.00 ^d^	1635.50 ± 8.50 ^b^	2091.50 ± 2.50 ^c^	1125.50 ± 16.50 ^d^	456.00 ± 6.00 ^c^
20	2586.50 ± 6.50 ^e^	1577.00 ± 8.00 ^c^	2016.50 ± 0.50 ^e^	1009.50 ± 14.50 ^e^	439.50 ± 7.50 ^d^
25	2521.50 ± 14.40 ^f^	1586.00 ± 10.00 ^c^	2003.00 ± 1.00 ^e^	935.50 ± 4.50 ^f^	417.00 ± 9.00 ^e^
30	2317.50 ± 10.50 ^g^	1502.00 ± 2.00 ^d^	1897.00 ± 9.00 ^f^	815.50 ± 8.50 ^g^	395.00 ± 7.00 ^f^

Values with a different letter in the same column are significantly different (*p* < 0.05). PV, TV and FV indicate the peak, trough and final viscosity of the starch granules, respectively. BD indicates the attenuation value of starch granules; SB indicates the retrogradation value of starch granules.

**Table 4 foods-14-02920-t004:** Effect of different levels of HACS addition at 40 Hz on rheological parameters in frequency sweep of HACS-UWRF blends.

HACS Addition/%	G′	G″	Tanδ
0	17.28 ± 0.10 ^f^	8.71 ± 0.03 ^c^	0.5 ± 0.00 ^ab^
5	18.18 ± 0.54 ^e^	8.66 ± 0.03 ^c^	0.48 ± 0.00 ^c^
10	19.32 ± 0.23 ^d^	8.77 ± 0.35 ^c^	0.45 ± 0.00 ^d^
15	19.81 ± 0.06 ^cd^	9.08 ± 0.23 ^c^	0.46 ± 0.01 ^d^
20	20.22 ± 0.13 ^c^	9.92 ± 0.09 ^b^	0.49 ± 0.00 ^bc^
25	21.24 ± 0.54 ^b^	10.80 ± 0.18 ^a^	0.51 ± 0.02 ^a^
30	23.00 ± 0.12 ^a^	11.19 ± 0.35 ^a^	0.49 ± 0.01 ^c^

Values with a different letter in the same column are significantly different (*p* < 0.05). G′, G″and tanδ indicate the elasticity and viscosity of the sample and its G″/G′ ratio, respectively.

**Table 5 foods-14-02920-t005:** Fourier transform infrared spectroscopy analysis of HACS-UWRF blends with HACS addition.

HACS Addition/%	R1047 cm^−1^/1022 cm^−1^	R1022 cm^−1^/995 cm^−1^
0	0.69 ± 0.00 ^e^	2.60 ± 0.00 ^a^
5	0.73 ± 0.01 ^d^	2.38 ± 0.00 ^b^
10	0.74 ± 0.00 ^d^	2.19 ± 0.00 ^c^
15	0.80 ± 0.00 ^c^	2.16 ± 0.00 ^d^
20	0.80 ± 0.00 ^c^	2.05 ± 0.00 ^e^
25	0.85 ± 0.01 ^b^	1.67 ± 0.00 ^f^
30	0.97 ± 0.04 ^a^	1.50 ± 0.03 ^g^

Values with a different letter in the same column are significantly different (*p* < 0.05). R1047 cm^−1^/1022 cm^−1^: the absorbance ratio of 1047 cm^−1^ and 1022 cm^−1^ representing the short-range orderliness of starch. R1022 cm^−1^/995 cm^−1^: the absorbance ratio of 1022 cm^−1^ and 995 cm^−1^ demonstrates the double helix content.

**Table 6 foods-14-02920-t006:** Effect of different levels of HACS addition on particle size parameters of HACS-UWRF blends.

HACS Addition/%	D10 (μm)	D50 (μm)	D90 (μm)	D (4,3) (μm)	D (3,2) (μm)
0	1.98 ± 0.01 ^c^	7.05 ± 0.03 ^d^	636.50 ± 6.34 ^b^	118.13 ± 2.37 ^b^	4.78 ± 0.02 ^d^
5	2.11 ± 0.02 ^a^	8.13 ± 0.12 ^a^	634.07 ± 7.54 ^b^	119.20 ± 2.72 ^ab^	5.24 ± 0.05 ^a^
10	2.04 ± 0.00 ^b^	7.36 ± 0.03 ^c^	640.13 ± 0.50 ^ab^	119.30 ± 0.20 ^ab^	4.92 ± 0.01 ^c^
15	2.06 ± 0.01 ^b^	7.52 ± 0.09 ^b^	636.20 ± 7.11 ^b^	119.37 ± 3.96 ^ab^	5.00 ± 0.05 ^b^
20	2.12 ± 0.01 ^a^	8.03 ± 0.06 ^a^	636.0 ± 9.66 ^b^	120.27 ± 3.77 ^ab^	5.25 ± 0.03 ^a^
25	2.06 ± 0.01 ^b^	7.36 ± 0.05 ^c^	642.73 ± 8.40 ^ab^	122.10 ± 3.55 ^ab^	4.95 ± 0.03 ^bc^
30	1.96 ± 0.01 ^d^	7.04 ± 0.04 ^d^	649.60 ± 1.85 ^a^	124.20 ± 0.92 ^a^	4.78 ± 0.02 ^d^

Values with a different letter in the same column are significantly different (*p* < 0.05). D10, D50, and D90 indicate particle sizes with a cumulative distribution of 10%, 50% and 90%, respectively. D (4,3) and D (3,2) indicate the mean diameter of starch granule volume and area, respectively.

## Data Availability

The original contributions presented in the study are included in the article/Appendix A, further inquiries can be directed to the corresponding authors.

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
