# Peer review of "Research on Physicochemical Properties and In Vitro Digestive Characteristics of High-Amylose Corn Starch–Ultrasound-Treated Waxy Rice Flour Blends"

_foods, 2025, doi:10.3390/foods14162920_

Round 1
Reviewer 1 Report
Comments and Suggestions for Authors
Line 31, please, replace “restricts the development of the glutinous rice products industry” with a specific, referenced statement about consumer health concerns (e.g., post‑prandial glycaemia).
Line 45, it would be beneficial to cite quantitative RS and eGI changes from Yang et al. (5) rather than only qualitative descriptors.
Line 60, please consider distinguishing clearly between “URWF” (your previous study) and “UWRF” (current acronym) to avoid typographical confusion.
Line 64, I recommend, adding 1–2 sentences explaining why HACS, rather than other slowly digestible starch sources, was chosen (e.g., availability, regulatory status).
Line 68, please, state an explicit hypothesis (e.g., “We hypothesised that progressive HACS substitution would…”) to help frame the experimental design.
Line 86, I recommend, specifying the ultrasound frequency (kHz) in addition to power (W).
Line 89, please, clarify whether %H refers to w/w dry basis; this affects reproducibility.
Line 100, it would be beneficial to state the rotation speed (rpm) of the magnetic stirrer during WSI/SP tests.
Line 112, please consider reporting DSC sample hermeticity (e.g., crucible sealing method) because water loss biases ΔH.
Line 120, please, describe RVA profile (heating/cooling steps) or cite AACC method number.
Line 126, it would be beneficial to declare the rheometer geometry (PP40, gap, etc.) and strain used during oscillatory scans to ensure tests were within the linear viscoelastic region.
Line 141, I recommend, including the step‑scan rate (° s⁻¹) for XRD to improve replicability.
Line 156, please, indicate dispersant medium temperature during particle‑size analysis; viscosity affects D(4,3).
Line 161, it would be beneficial to correct the buffer description: sodium acetate at pH 5.2 is typically 0.1 mol L⁻¹, not 0.2 mol L⁻¹; verify.
Line 168, please, justify the ethanol quenching step volume (ratio to aliquot) and confirm that centrifugation pellets were discarded.
Line 183, I recommend, specifying the post‑ANOVA mean‑separation test (e.g., Tukey) used to assign superscripts.
Results & Discussion
Line 194, please consider presenting amylose, WSI and SP values as mean ± SD with CV% in Table 1 to contextualise variability.
Line 213, please, double‑check DSC ΔH units; reported values of 316–722 J g⁻¹ exceed typical starch gelatinisation enthalpy (5–20 J g⁻¹); likely a unit or baseline integration error.
Line 241, it would be beneficial to relate PV decrease to specific industrial processing implications (e.g., extrusion viscosity).
Line 254, I recommend, discussing whether reduced setback (SB) correlates with lower retrogradation during storage; cite shelf‑life studies if available.
Line 270, please, correct “tanδ initially decreases and then rises… increases viscosity” — viscosity does not increase when both moduli fall; rephrase for clarity.
Line 291, please consider adding Williams–Landel–Ferry (WLF) or power‑law modelling to strengthen rheological interpretation.
Line 321, I recommend, correlating FTIR R₁₀₄₇/₁₀₂₂ trends with XRD crystallinity (Fig. 6b) quantitatively (e.g., Pearson r).
Line 349, please, include crystallinity error bars and statistical significance in Fig. 6b.
Line 392, I recommend, discussing whether the modest increase in D(4,3) (118 → 124 µm) is practically relevant to milling or hydration behaviour.
Line 406, please, present HI and eGI values with SD and indicate whether the in‑vitro method is validated against in‑vivo GI.
Line 409, it would be beneficial to cite enzyme kinetic limitations (α‑amylase vs amyloglucosidase) when interpreting RS changes.
Line 423, please consider tempering the conclusion “effectively improved” with quantitative statements (e.g., PV decreased by 32 %, RS increased by 16 %)
Line 66, I recommend, explicitly contrasting your ultrasound‑HACS approach with chemical‑modification routes (e.g., octenyl succinate) to demonstrate novelty.
Line 425, please consider adding a limitations paragraph: (i) lack of sensory evaluation of rice cakes; (ii) absence of in‑vivo glycaemic testing; (iii) scalability of ultrasound treatment for industrial flour volumes.
Author Response
Dear Editor and Reviewers:
Thank you very much for your valuable comments on our manuscript (foods-3768322) entitled "Research on physicochemical properties and in vitro digestive characteristics of high amylose corn starch-ultrasound treated waxy rice flour blends ".
We have responded to all the questions put forward by the reviewers with answers or explanations, and have revised our paper accordingly. The corrections were shown in red. We hope this version would meet with your approval. But, we are willing to do further revision if there is something deficiency.
We thank the reviewers for their valuable advice, and are grateful to you for all the favor you have done for us.
Sincerely yours,
Yuxing Wang
The following are our answers to the questions put forward by the reviewer:
Response to Reviewer 1.
Question 1: Line 31, please, replace “restricts the development of the glutinous rice products industry” with a specific, referenced statement about consumer health concerns (e.g., postprandial glycaemia).
Answer 1: Thank you for your valuable advice. We have supplemented and revised the relevant content in the Introduction. The specific modifications are as follows: "Thus, it is not suitable for people with high blood sugar or related sub-health groups" (Page 1 Line 37). I think this revision aims to be more in line with the background of health research.
Question 2: Line 45, it would be beneficial to cite quantitative RS and eGI changes from Yang et al. (5) rather than only qualitative descriptors.
Answer 2: Thank you for your valuable advice. Your suggestion on supplementing the quantitative description of RS and eGI changes is very pertinent. We reviewed the research results of Yang et al. And supplemented the relevant quantitative data, modifying it as: "The content of amylose significantly increased from 0.51% to 1.91% (P<0.05) and the content of resistant starch (RS) significantly increased from 4.03% to 7.69% (P<0.05)" (Page 2 Line 50-52). This modification replaces the original qualitative description with specific numerical values and statistical significance results, making the expression more precise.
Question 3: Line 60, please consider distinguishing clearly between “URWF” (your previous study) and “UWRF” (current acronym) to avoid typographical confusion.
Answer 3: Thank you for your careful inspection and valuable suggestions. We standardized the relevant abbreviations as "UWRF" (ultrasound-treated waxy rice flour), and reviewed the full text to ensure that all expressions related to this term are consistent and avoid confusion in typesetting and understanding.
Question 4: Line 64, I recommend, adding 1–2 sentences explaining why HACS, rather than other slowly digestible starch sources, was chosen (e.g., availability, regulatory status).
Answer 4: Thank you for your professional suggestions. We have revised the original text as "The amylose content in high amylose corn starch (HACS) is greater than 70%. Compared with common starch, HACS exhibits stronger heat resistance. After high-temperature treatment, more RS components are retained, thereby conferring higher anti-digestive properties and a lower GI. In recent years, HACS has been widely used in food preparation due to these advantages." (Page 2 Line 64-68). this modification highlights the advantages of HACS in heat resistance, digestive resistance and practical application by comparing with common starch. Make the reasons for choosing HACS more sufficient and the logic clearer.
Question 5: Line 68, please, state an explicit hypothesis (e.g., “We hypothesised that progressive HACS substitution would…”) to help frame the experimental design.
Answer 5: Thank you for your valuable suggestions. We have added more explicit assumptions in the Introduction. The specific content has been revised as follows: "Although there are studies on adding HACS to food, few reports have focused on its application in ultrasound-treated foods. We hypothesized that combining these two physical methods (ultrasound and HACS compounding) would help explore changes in the physicochemical properties and in vitro digestibility of the blends." (Page 3 Line 77-80). This hypothesis clarifies the direction of the synergistic effect between "progressive HACS substitution" and "ultrasonic treatment", and at the same time leads to the content of studying the physical and chemical properties and digestive characteristics of HACS substitution in ultrasonically treated foods.
Question 6: Line 86, I recommend, specifying the ultrasound frequency (kHz) in addition to power (W).
Answer 6: Thank you for your attention to the details of the experiment and your valuable suggestions, which have helped us improve the description of the experimental method. We supplemented the relevant content in 2.2. Pre-treatment of raw materials, modifying it tothe ultrasonic conditions were as follows: "frequency 25 kHz, time 4 min, power 400 W, and drying temperature 49 °C."
Question 7: Line 89, please, clarify whether %H refers to w/w dry basis; this affects reproducibility.
Answer 7: Thank you for your attention to the details of the experimental conditions. %H represents the addition ratio of HACS, specifically referring to the percentage of the mass of HACS to the dry basis weight of UWRF (w/w). We have supplemented HACS in 2.2. Pre-treatment of raw materials: "HACS was added to UWRF at a ratios of 0%, 5%, 10%, 15%, 20%, 25%, and 30% (w/w, based on the dry weight of UWRF) and thoroughly mixed". I think this modification can facilitate other researchers to reproduce the experiment.
Question 8: Line 100, it would be beneficial to state the rotation speed (rpm) of the magnetic stirrer during WSI/SP tests.
Answer 8: Thank you for your attention to the details of the experiment. we supplemented that during the WSI/SP test, the rotational speed of the magnetic stirrer was 160 r/min. (Page 5 Line 125)
Question 9: Line 112, please consider reporting DSC sample hermeticity (e.g., crucible sealing method) because water loss biases ΔH.
Answer 9: Thank you for your attention to the details of the experiment. We have added the content in 2.5.DSC of HACS-UWRF blends as follows: "The crucible was compacted and sealed, and equilibrated at 4°C in a refrigerator".
Question 10: Line 120, please, describe RVA profile (heating/cooling steps) or cite AACC method number.
Answer 10: Thank you for your professional suggestions, which have helped us refine the detailed description of the experimental method. We add in the article: "The RVA program was set as follows: equilibration at 50 °C for 1 min, then heat at a rate of 6°C/min to the target temperature of 95°C and maintain for 5 min, then cool at the same rate to 50 °C and maintain for 2 min. The gelatinization curve of the composite powder was then recorded". (Page 6 Line 148-151)
Question 11: Line 126, it would be beneficial to declare the rheometer geometry (PP40, gap, etc.) and strain used during oscillatory scans to ensure tests were within the linear viscoelastic region.
Answer 11: Thank you for your meticulous attention to the details of the experiment. We have supplemented the detailed parameters of the static/dynamic rheology test, specifically as follows: "Take a certain amount of emulsion and quickly transfer it onto a preheated ø 40mm plate, and seal it with silicone oil. Set the spacing at 1mm and the strain at 0.5% (in the linear viscoelastic zone). Test procedure: After the emulsion was equilibrated at 25°C, for 3 min, frequency was scanned from 0.1 to 100 Hz". (Page 6 Line 157-160)
Question 12: Line 141, I recommend, including the stepscan rate (° s⁻¹) for XRD to improve replicability.
Answer 12: Thank you for your meticulous attention to the details of the experiment. the XRD test conditions we conducted at that time were: Cu radiation, voltage: 40 kV, current: 150 mA, scanning range: 5° to 40°, step size: 0.02°, scanning speed: 6°/min, and measurement of the 2θ X-ray diffraction spectrum. Regarding the scanning rate (° s⁻¹) you mentioned, Our understanding is that the scanning speed of 6°/min" can be converted to "0.1/s" (Page 7 Line 177), which is actually consistent with the "° s⁻¹" unit you suggested. Therefore, no further additions or modifications will be made in the text.
Question 13: Line 156, please, indicate dispersant medium temperature during particlesize analysis; viscosity affects D(4,3).
Answer 13: Thank you for your attention to the details of the experiment. When testing the particle size of mixed powders, we adopt wet testing and select deionized water as the dispersant. Therefore, we believe that its temperature has little influence on the viscosity of the particle size. However, in order to comply with the standardization of the experimental method, we once again supplement in the method as: "mix the sample with deionized water to prepare a 1% emulsion". (Page 7 Line 190)
Question 14: Line 161, it would be beneficial to correct the buffer description: sodium acetate at pH 5.2 is typically 0.1 mol L⁻¹, not 0.2 mol L⁻¹; verify.
Answer 14: Thank you for your meticulous attention to the details of the experiment. Regarding the issue of buffer concentration, we have re-verified the experimental records and confirmed that the concentration of sodium acetate buffer used in the in vitro digestion experiment in this study was 0.2 mol L⁻¹ (pH 5.2). This concentration setting is based on the digestion system parameters optimized in the laboratory in the early stage, and the relevant concentrations have been verified for their applicability in the pre-experiments.
Question 15: Line 168, please, justify the ethanol quenching step volume (ratio to aliquot) and confirm that centrifugation pellets were discarded.
Answer 15: Thank you for your suggestion. We have supplemented the experimental procedures of 2.12. In vitro digestion of HACS-UWRF blends, including the ethanol addition amount you mentioned, the treatment of the precipitate after centrifugation, and the DNS method for determining glucose. The specific modification is (1) "4mL of ethanol was added to inactivate the enzyme, and the tubes were centrifuged at 10,000 r/min and 4°C for 5 min. The precipitate was discarded". (2) "DNS reagent (2 mL) was added to dilution solution (1 mL), and the mixture was heated in a boiling water bath for 5 min. After cooling and further dilution, the absorbance was measured at 520 nm. The glucose content was calculated through the glucose standard curve (y=0.9765x+0.0383, R2=0.9992)". Once again, thank you for your professional guidance, which has helped us further improve the standardized description of the experimental method.
Question 16: Line 183, I recommend, specifying the postANOVA meanseparation test (e.g., Tukey) used to assign superscripts.
Answer 16: Thank you for your suggestion. We further modified it in 2.13.Statistical analysis. The revised content is: "statistical analysis was performed using IBM SPSS 19.0, including one-way analysis of variance (ANOVA) and Duncan’s multiple range test. Differences were considered significant at P < 0.05, and results were expressed as mean ± standard deviation".
Question 17: Line 194, please consider presenting amylose, WSI and SP values as mean ± SD with CV% in Table 1 to contextualise variability.
Answer 17: Thank you for your suggestions. In the analysis section, we supplemented the quantitative descriptions of amylose, WSI and SP. Such as "the amylose content in the blends significantly increased from 3.82% to 16.78% (P<0.05). And the WSI significantly decreased from 22.9% to 6.17% (P<0.05), while the SP also decreased significantly from 19.63% to 14.02% (P<0.05) "(Page 9 Line 236,240). This makes the dynamic changes of the data more intuitive. Furthermore, we have marked ± SD in Table 1, and marked the significance with different lowercase letters (P< 0.05). Therefore, we believe that the current presentation method can clearly reflect the differences in the data, so we do not consider adding CV% for the time being.
Question 18: Line 213, please, doublecheck DSC ΔH units; reported values of 316–722 J g⁻¹ exceed typical starch gelatinisation enthalpy (5–20 J g⁻¹); likely a unit or baseline integration error.
Answer 18: Thank you for your meticulous attention to the experimental data. After repeated confirmation, it was found that the problem originated from an incorrect baseline integration method during the data processing, which led to deviations in the results. At present, we have re-analyzed and recalculated the original DSC data, corrected the processing method, and updated the correct ΔH (14.82–18.34 J g⁻¹) (Table 2; Page 10 Line 261). Thank you again for your strict correction.
Question 19: Line 241, it would be beneficial to relate PV decrease to specific industrial processing implications (e.g., extrusion viscosity).
Answer 19: Thank you for your suggestions. We have made targeted supplements in the gelatinization analysis of the HACS-UWRF blends (Page 11 Line 277; Page 12 Line 290): Firstly, we have added a quantitative analysis of the PV: "PV significantly decreases from 3386 mPa·s to 2317.5 mPa·s (P < 0.05)", making the change in PV values more intuitive; Furthermore, we explained the mechanism by which HACS substitution leads to a decrease in PV in combination with relevant literature (Reference 33-36). Such as: "The addition of HACS enhances the rigidity of starch granules, facilitating the formation of a spatial barrier effect during gelatinization and inhibiting the water absorption and expansion capacity of starch granules [34], which is consistent with the hydration property results." Finally, we also further explain the application value of this low-viscosity characteristic mixed powder in industrial processing, for example: "This low-viscosity and thermally stable starch property has better adaptability in the production and processing of products such as cakes and bread". In conclusion, we believe that these supplementary contents further enhance the rigor and practical value of the article.
Question 20: Line 254, I recommend, discussing whether reduced setback (SB) correlates with lower retrogradation during storage; cite shelflife studies if available.
Answer 20: Thank you for your professional advice. We have supplemented the relationship between the reduction of the setback (SB) and the quality of food products, specifically modifying it as "SB significantly decreased from 606 mPa·s to 395 mPa·s (P < 0.05), indicating that the blends are less prone to regeneration and hardening during temperature reduction, and more likely to form a gel with higher strength and better elasticity after cooling. Therefore, it exhibits enhanced edible quality and an extended shelf life" (Page 12 Line 294-297). And we supplemented relevant research literature on shelf life as support (Reference 37), making the scientific and practical significance of this part more substantial.
Question 21: Line 270, please, correct “tanδ initially decreases and then rises… increases viscosity” — viscosity does not increase when both moduli fall; rephrase for clarity.
Answer 21: Thank you for your valuable suggestions. Regarding 3.4. Effect of HACS addition on the rheological properties of HACS-UWRF, we have made the following modifications: Firstly, the frequency scan analysis in the original determination of dynamic viscoelastic properties was retained, with a focus on analyzing the influence of the addition amount of HACS on the G' , G ", and tanδ in the frequency scan of the HACS-UWRF blends. As for the temperature change part, since it was found that its analysis conclusion was highly consistent with the frequency scan results, in order to avoid repetition and redundancy, its core information has been integrated into the frequency scan analysis. And further language modifications were made to this part of the content, enhancing the accuracy and readability of the article.The relevant modified contents have been prominently presented in page 13, line 303-317 of the article.
Question 22: Line 291, please consider adding Williams–Landel–Ferry (WLF) or powerlaw modelling to strengthen rheological interpretation.
Answer 22: Thank you for your valuable suggestions. In the article, we have added an analysis of the influence of HACS addition amount on the apparent viscosity of the blends, further analyzing the characteristics of the mixed powder from the perspective of static rheology. Finally, based on the comprehensive analysis of dynamic and static (Answer 21) rheological properties, "This indicates that the starch paste is mainly elastic with weakened viscosity, belonging to a typical elastic solid weak gel system" (Page 13 Line 307), "Combining dynamic and static rheological properties, it is found that HACS addition increases the shear-thinning degree and gel strength of the starch paste while enhancing starch structural stability, consistent with the gelatinization characteristic results." (Page 15 Line 336).
Therefore, after careful consideration, we believe that the existing experimental data can fully support the core conclusion of rheological properties and are highly consistent with the overall framework and objectives of the research. so we have decided not to add any additional Williams-Landel-Ferry (WLF) or power-law models. Thank you again for your professional guidance. Your suggestions have helped us review this part more comprehensively.
Question 23: Line 321, I recommend, correlating FTIR R₁₀₄₇/₁₀₂₂ trends with XRD crystallinity (Fig. 6b) quantitatively (e.g., Pearson r).
Answer 23: Thank you for your professional advice. In combination with relevant literature, in the analyses of FTIR in section 3.5 and XRD in Section 3.6, we have focused on discussing the influence of the addition of HACS on the starch structure in the blend system, and directly correlated the change of relative crystallinity in XRD with the ratios of R1022 cm-1/995 cm-1 and R1022 cm-1/995 cm-1 in FTIR. Make the logic of this part clearer. The relevant modified contents have been prominently presented in page 15 to 17 of the article.
Question 24: Line 349, please, include crystallinity error bars and statistical significance in Fig. 6b.
Answer 24: Thank you for your attention to the details of the experiment. We have re-standardized and added error bars in Fig. 6b. to reflect the degree of data dispersion, and added a caption "Values with different letters in the graph are significantly different (P< 0.05)".
Question 25: Line 392, I recommend, discussing whether the modest increase in D(4,3) (118 → 124 µm) is practically relevant to milling or hydration behaviour.
Answer 25: Thank you for your suggestion. We reorganized this part of the logic, combining the phenomenon that the addition of HACS led to an increase in D (4,3) in the mixed powder with the previously mentioned results such as the decrease in water WSI and SP, as well as the more compact aggregation of starch particles observed by SEM. The specific modification is: "the average particle size of the blends gradually increased from 118.133μm to 124.2μm, consistent with the phenomenon observed by SEM. This increase in particle size and change in aggregation state reduce the contact area between particles and water, inhibiting the permeation of water molecules and the dissolution of starch chains, ultimately resulting in a significant decrease in WSI and SP (P<0.05)." (Page 20 Line 425-428). This systematically expounds the intrinsic connections among these experimental phenomena, further enhancing the correlation and completeness of the experimental data.
Question 26: Line 406, please, present HI and eGI values with SD and indicate whether the invitro method is validated against invivo GI.
Answer 26: Thank you for your suggestions. We standardize the HI and eGI values in the form of SD. We elaborated in detail on the connection and differences between this method and in vivo GI determination, and cited relevant literature for support (Reference 62). The specific modification is as follows: "Previous studies have confirmed that HI and eGI values obtained via in vitro simulation methods are highly correlated with the actual in vivo GI in humans. However, in vitro models cannot fully replicate in vivo dynamic processes (e.g., hormone regulation, the role of gut microbiota), so eGI results require further verification through animal experiments or human trials". (Page 21 Line 451-455)
Question 27: Line 409, it would be beneficial to cite enzyme kinetic limitations (αamylase vs amyloglucosidase) when interpreting RS changes.
Answer 27: Thank you for your professional advice. We have revised this part of the content:"The increase of RS inhibits the activities of α-amylase and amyloglucosidase, effectively stabilizing postprandial blood glucose fluctuations."(Page 21 Line 444)
Question 28: Line 423, please consider tempering the conclusion “effectively improved” with quantitative statements (e.g., PV decreased by 32 %, RS increased by 16 %)
Answer 28: Thank you for your suggestion. We supplemented the specific data on RS changes: "while the RS content significantly increased from 53.57%±0.65 to 63.83%±0.53 (P < 0.05) "(Page 21 Line 442); "PV significantly decreases from 3386 mPa·s to 2317.5 mPa·s (P < 0.05)" (Page 11 Line 277). We comprehensively examined similar descriptions throughout the text and further supplemented the corresponding quantitative data to make the presentation of the experimental results more convincing.
Question 29: Line 66, I recommend, explicitly contrasting your ultrasoundHACS approach with chemicalmodification routes (e.g., octenyl succinate) to demonstrate novelty.
Answer 29: Thank you for your suggestion. We supplement this part in the introduction, and the specific modification is: "Most current research focuses on improving the functional properties of starch through physical, chemical or biological means. However, due to issues such as residual chemical reagents, complex processes and high costs, the application of chemical modification and enzymatic modification methods in the food processing field has been limited to a certain extent. In contrast, physical modification techniques do not require exogenous chemical substances, are easy to operate, and have high safety, which better meets the green development needs of the food industry. " (Page 2 Line 41-48). However, the single ultrasonic effect is limited in reducing starch digestibility. HACS have heat resistance and anti-digestion properties. Therefore, we further proposed the hypothesis of the synergistic effect of two physical methods, namely ultrasonic action and the combined addition of HACS, and studied the variation patterns of their influences on the physicochemical properties of glutinous rice flour and the digestion characteristics of rice cakes. This is done to better demonstrate the novelty and practical value of the work in this paper, making the logic of the research background and innovation points clearer.
Question 30: Line 425, please consider adding a limitations paragraph: (i) lack of sensory evaluation of rice cakes; (ii) absence of invivo glycaemic testing; (iii) scalability of ultrasound treatment for industrial flour volumes.
Answer 30: Thank you for your suggestions. We have added restricted paragraphs. (Page 23 Line 479)

Reviewer 2 Report
Comments and Suggestions for Authors
The research is dealing with influence of addition of HAS on rice flour properties. The novelty and scientific contribution of the research is not visible in the present form, and if there is one, it should be better presented (flour itfels is not sufficient novelty).
Other, more specific comments are as follows:
In the introduction, please specify where gloutnous rice products are popular, it is linked to the regions, you cannot claim it as general rule for the whole World.
line 34: "hot topic in high-value development." - should it be "in high-value product development"?
line 96: "1mlo/L" - please, correct the unit
line 97: "dilute (100 mL)" - dilute with what? 100 mL was the final volume or the volume of the solvent for dilution?
Add the equation for calculating amylose content and elaborate how does (not) amylopectin interfere with the result
line 104-105: "supernatant and precipitate were respectively dried and weighed." Temperature of drying? For specific period of time or until constant mass?
precipitate was dried, but wet mass was taken in the calculation? why was it dried?
DSC? There was no cooling of the sample?
conditions of RVA testing?
line 133: FTIR was done "on rice cake", and the title says "HACS-UWRF blends"
2.12. In vitro digestion of HACS-UWRF blend - it is not clear how the glucose content was quantified
line 185: "The experimental data were statistically analyzed using IBM SPSS 19.0 software at a significant level of P<0.05." Which test(s) was/were used?
when writing discussion, combine the results of different analyses that you have conducted in the research, they can be easily and nicely combined and one propeerty is reflected in the results of the other ones.
Author Response
Dear Editor and Reviewers:
Thank you very much for your valuable comments on our manuscript (foods-3768322) entitled "Research on physicochemical properties and in vitro digestive characteristics of high amylose corn starch-ultrasound treated waxy rice flour blends ".
We have responded to all the questions put forward by the reviewers with answers or explanations, and have revised our paper accordingly. The corrections were shown in red. We hope this version would meet with your approval. But, we are willing to do further revision if there is something deficiency.
We thank the reviewers for their valuable advice, and are grateful to you for all the favor you have done for us.
Sincerely yours,
Yuxing Wang
The following are our answers to the questions put forward by the reviewers:
Response to Reviewer 2.
Question 1: The research is dealing with influence of addition of HAS on rice flour properties. The novelty and scientific contribution of the research is not visible in the present form, and if there is one, it should be better presented (flour itfels is not sufficient novelty).
Answer 1: Thank you for your suggestion. Firstly, in the introduction, we supplemented the advantages of physical ultrasonic modification over other chemical and biological modifications, and the specific modification is: "Most current research focuses on improving the functional properties of starch through physical, chemical or biological means. However, due to issues such as residual chemical reagents, complex processes and high costs, the application of chemical modification and enzymatic modification methods in the food processing field has been limited to a certain extent. In contrast, physical modification techniques do not require exogenous chemical substances, are easy to operate, and have high safety, which better meets the green development needs of the food industry." (Page 2 Line 41-48). However, single ultrasound has limited effect on reducing starch digestibility, so new methods need to be explored.
Then, by comparing HACS with common starch, we further highlighted the advantages of HACS in terms of heat resistance, digestion resistance and practical application. The revised content is:"The amylose content in high amylose corn starch (HACS) is greater than 70%. Compared with common starch, HACS exhibits stronger heat resistance. After high-temperature treatment, more RS components are retained, thereby conferring higher anti-digestive properties and a lower GI. In recent years, HACS has been widely used in food preparation due to these advantages." (Page 2 Line 64-68). This makes our reasons for choosing HACS more solid and the logic clearer. Next, we further proposed the hypothesis of the synergistic effect of two physical methods, namely ultrasonic action and the combined addition of HACS, and studied the variation patterns of their influences on the physicochemical properties of glutinous rice flour and the digestion characteristics of rice cakes. This is done to better demonstrate the novelty and practical value of the work in this paper, making the logic of the research background and innovation points clearer.
Question 2: In the introduction, please specify where gloutnous rice products are popular, it is linked to the regions, you cannot claim it as general rule for the whole World.
Answer 2: Thank you very much for your suggestion. We have made modifications to this part. The current statement is: "Its products are highly popular among consumers in Southeast Asia". (Page 1 Line 35)
Question 3: line 34: "hot topic in high-value development." - should it be "in high-value product development"?
Answer 3: Thank you for your attention to the details of the paper. We have corrected "hot topic in high-value development." to "in high-value product development" to make the meaning more explicit. (Page 2 Line 39)
Question 4: line 96: "1mlo/L" - please, correct the unit
Answer 4: Thank you for your attention to the details of the experiment. We have immediately corrected "1mol/L". At the same time, all the units involved in the entire text were checked to ensure that similar clerical errors had been thoroughly corrected.
Question 5: line 97: "dilute (100 mL)" - dilute with what? 100 mL was the final volume or the volume of the solvent for dilution?
Answer 5: Thank you for your detailed question. This step involves diluting the sample solution to a 100 mL volumetric flask with distilled water. Furthermore, in response to your several questions regarding the determination of Amylose, we have supplemented the method for the determination of amylose again in 2.3. Amylose content of HACS-UWRF blends to make the description of the experimental method more accurate.
Question 6: Add the equation for calculating amylose content and elaborate how does (not) amylopectin interfere with the result
Answer 6: Thank you for your professional suggestions. Our research group referred to the Chinese national standard GB/T 15683-2008 "Determination of Amylose Content in Rice". The experiment was conducted by measuring the absorbance of the sample at 720nm (based on the specific color reaction of the complex formed by amylose and iodine), and substituting it into the standard curve equation (y=ax + b) to calculate the amylose content. The specific data of this standard curve (y=0.0054x+0.0887,R2=0.9928) (Page 4 Line 111-120)
Furthermore, regarding the interference issue of amylopectin you mentioned, we will further explain: The core principle of this method is to take advantage of the high affinity between amylose and iodine to form a blue complex, while amylopectin has a relatively weak binding ability with iodine (only forming a purple-red complex). When measured at a wavelength of 720 nm, the absorbance contribution of the amylopectin - iodine complex is extremely low, which can minimize its interference. The influence on the experimental results is negligible within the allowable error range, so no additional correction is required.
Question 7: line 104-105: "supernatant and precipitate were respectively dried and weighed." Temperature of drying? For specific period of time or until constant mass?
precipitate was dried, but wet mass was taken in the calculation? why was it dried?
DSC? There was no cooling of the sample?
Answer 7: Thank you for the questions you raised about the details of the experiment. Regarding the questions about the drying conditions for the determination of WSI and SP and the DSC operation process, we provide supplementary explanations based on the test records as follows:
- Drying temperature: The supernatant of the sample after centrifugation is poured into an aluminum box and dried in an oven at 105℃ until a constant weight is achieved (usually taking 4-6h). During this period, the mass difference between two measurements is confirmed to be ≤ 0.001g through multiple weighing. At this point, the sample is taken out and the dry weight (W2) is recorded.(Page 5 Line 127)
- The precipitate in the centrifuge tube was not dried and the wet weight (W3) was directly weighed. Since SP reflects the state of starch particles after absorbing water and swelling, the wet weight can more accurately reflect their swelling capacity.(Page 5 Line 128)
- DSC operation process: When preparing the sample, add deionized water at a ratio of 1:2 (w/v), seal it in a crucible, and then balance it overnight in a 4℃ refrigerator. Therefore, the sample has been balanced to a stable state before heating, ensuring the reliability of the test data. During the testing phase, the emulsion sample was heated from 80℃ to 120℃ in a nitrogen environment to obtain the DSC curve. It is hoped that these supplements can further clarify the key details of the experimental operation.(Page 5 Line 137)
Question 8: conditions of RVA testing?
Answer 8: Thank you for your professional suggestions, which have helped us refine the detailed description of the experimental method. We add in the article: "The RVA program was set as follows: equilibration at 50 °C for 1 min, then heat at a rate of 6°C/min to the target temperature of 95°C and maintain for 5 min, then cool at the same rate to 50 °C and maintain for 2 min. The gelatinization curve of the composite powder was then recorded". (Page 6 Line 148-150)
Question 9: line 133: FTIR was done "on rice cake", and the title says "HACS-UWRF blends"
Answer 9: Thank you for your careful check and valuable corrections. It has been confirmed that the title "HACS-UWRF Mixture" is accurate. In article 2.8. FTIR of HACS-UWRF blends, the object measured by FTIR was wrongly written as "rice cake", which is a clerical error. Therefore, we have made unified corrections to the relevant content to ensure that all descriptions are clear and avoid information confusion. Thank you again from the bottom of our hearts for helping us point out this oversight.
Question 10: 2.12. In vitro digestion of HACS-UWRF blend - it is not clear how the glucose content was quantified.
Answer 10: Thank you for your suggestion. We supplemented the experimental steps for determining the DNS of glucose in 2.12. In vitro digestion of HACS-UWRF blends. The specific modification is: "DNS reagent (2 mL) was added to dilution solution (1 mL), and the mixture was heated in a boiling water bath for 5 min. After cooling and further dilution, the absorbance was measured at 520 nm. The glucose content was calculated through the glucose standard curve (y=0.9765x+0.0383, R2=0.9992)". Once again, thank you for your professional guidance, which has helped us further improve the standardized description of the experimental method.
Question 11: line 185: "The experimental data were statistically analyzed using IBM SPSS 19.0 software at a significant level of P<0.05." Which test(s) was/were used?
Answer 11: Thank you for your suggestion. We further modified it in 2.13.Statistical analysis. The revised content is: "statistical analysis was performed using IBM SPSS 19.0, including one-way analysis of variance (ANOVA) and Duncan’s multiple range test. Differences were considered significant at P < 0.05, and results were expressed as mean ± standard deviation".
Question 12: when writing discussion, combine the results of different analyses that you have conducted in the research, they can be easily and nicely combined and one propeerty is reflected in the results of the other ones.
Answer 12: Thank you for your valuable suggestions. Your raising of this question is highly inspiring for enhancing the integrity and logic of the article. We further conducted in-depth analysis and discussion on the experimental results of each part of the article, including closely integrating the experimental results of amylose content, WSI, SP, gelatinization characteristics, thermal properties, structural properties (XRD and FTIR), SEM and In vitro digestibility, etc., so that one attribute can be reflected in the results of other attributes. The corrections were shown in red.
In addition, based on the results of the previous integrated analysis, we have also made corresponding modifications to the conclusion section. The specific modifications are as follows: "The addition of HACS significantly affected the physicochemical properties and in vitro digestibility of the HACS-UWRF blend system. With increasing HACS content, amylose content significantly increased (P < 0.05), which compacted the internal structure of starch molecules, gradually increased the average particle size, and smoothed the surface of starch particles. Meanwhile, HACS alleviated the rupture of starch particles in the blends, inhibiting starch particle water absorption and expansion. This reduced the gelatinization viscosity of the mixed powder, ultimately enhancing the thermal paste stability and shear resistance of the starch and suppressing starch retrogradation and regeneration. XRD and FTIR results further indicated that all blends exhibited typical A-type crystallization. HACS addition enhanced the RC and short-range ordered structure of starch. Rheological results showed that the starch paste of the blends is a pseudoplastic non-Newtonian fluid, exhibiting shear-thinning behavior with the system being mainly elastic. In vitro digestibility results revealed that with increasing HACS content, the RS content of rice cakes significantly increased from 53.57%±0.65 to 63.83%±0.53 (P < 0.05), and the eGI value significantly decreased from 69.16±0.15 to 58.97±0.01 (P < 0.05). Rice cakes belong to medium-GI foods. This research can be applied to develop glutinous rice starch foods with slow-glycemic increase properties."

Reviewer 3 Report
Comments and Suggestions for Authors
Foods (ISSN 2304-8158)
Manuscript ID foods-3768322
Title: Research on physicochemical properties and in vitro digestive characteristics of high amylose corn starch-ultrasound treated waxy rice flour blends
The manuscript addresses an interesting and timely topic — the modification of starch properties through the incorporation of high amylose corn starch (HACS) into ultrasound-treated waxy rice flour (UWRF) to improve its nutritional profile and physicochemical behavior. The application of this composite for the development of slow-glycemic rice-based foods is of potential relevance in the fields of food science, nutrition, and functional foods.
However, the manuscript in its current form has significant limitations in its scientific clarity, structure, language, and depth of discussion. Before being considered for publication in Foods, major revisions are necessary.
1.Language and Style (Major Revision Required)
-The manuscript suffers from poor English usage, inconsistent tense, awkward phrasing, and non-standard scientific expression, which makes it difficult to follow in many parts.
-Examples include:
"This contained that amylose inhibits expansion..." → grammatically incorrect and unclear.
ΔH indicates the enthalpy change." → overly simplistic and redundant.
-Numerous run-on sentences, unclear subject-verb agreements, and imprecise technical descriptions weaken the scientific impact.
-A thorough language and technical editing by a native English speaker with expertise in food science is urgently needed.
- Scientific Rationale and Objectives
-The introduction is missing from the content provided. However, the abstract and body lack a strong hypothesis-driven framework.
-There is no clear explanation of why ultrasound treatment was combined with HACS addition — two distinct processing techniques. What is the rationale for this dual approach?
-It is unclear whether the changes observed are due to ultrasound treatment, HACS, or a synergistic interaction. Control experiments using only HACS and only ultrasound-treated WR flour are needed to separate effects.
- Data Interpretation and Scientific Depth
-The manuscript primarily reports trends (e.g., amylose content increased, viscosity decreased) but lacks mechanistic insight and does not connect data across sections.
-Statements like "This may be attributed to..." or "This is because..." are not always supported with evidence (e.g., microscopy, FTIR, or crystallography) or citations.
- Conclusions
-While the results suggest changes in starch behavior and digestibility, the claims made in the conclusion are overly broad and unsupported by strong mechanistic evidence.
-For example, "This product can be applied to glutinous rice food..." is a commercial implication that should not be drawn without sensory testing, shelf-life studies, and consumer acceptance analysis.
-The reference to “co-gel effect” is vague — further clarification or quantification (e.g., via rheology) is needed.
Minor Issues and Specific Comments:
- Line 13: "solubility and swelling power gradually decreased, while the average particle size of the blends enlarges" → “enlarges” should be “increased” or “became larger.”
- Line 215: "which makes the blends less capable of combining with water" → consider replacing with “which reduces the water absorption capacity.”
- Line 242: “a reduction of 31.56%” → not scientifically meaningful without SD or error values.
- Line 408: "thus rendering the rice cake a medium-GI food" → Should be supported by comparison to reference GI categories.
- Line 424: “mixed powder can promote the formation of a more effective co-gel effect…” → Unclear and vague.
Recommendations for Improvement:
- Rewrite the manuscript with professional English editing to enhance clarity and scientific rigor.
- Strengthen the introduction and rationale, explaining the novelty and why ultrasound + HACS is investigated together.
- Include or clarify the experimental methodology in detail, especially regarding ultrasound treatment and digestibility assays.
- Clearly distinguish effects due to ultrasound vs. HACS vs. their interaction.
- Provide comprehensive statistical analysis, include error bars, and define significance.
- Improve mechanistic discussion using supportive literature and techniques (e.g., FTIR, XRD, microscopy).
- Temper the conclusions — suggest potential applications without overpromising or extrapolating beyond data.
Recommendation:
Major Revision
The topic is relevant and potentially valuable, but the current manuscript needs substantial rewriting, deeper analysis, and clearer presentation before it is suitable for publication.
Comments on the Quality of English Language1.Language and Style (Major Revision Required)
-The manuscript suffers from poor English usage, inconsistent tense, awkward phrasing, and non-standard scientific expression, which makes it difficult to follow in many parts.
-Examples include:
"This contained that amylose inhibits expansion..." → grammatically incorrect and unclear.
ΔH indicates the enthalpy change." → overly simplistic and redundant.
-Numerous run-on sentences, unclear subject-verb agreements, and imprecise technical descriptions weaken the scientific impact.
-A thorough language and technical editing by a native English speaker with expertise in food science is urgently needed.
Author Response
Dear Editor and Reviewers:
Thank you very much for your valuable comments on our manuscript (foods-3768322) entitled "Research on physicochemical properties and in vitro digestive characteristics of high amylose corn starch-ultrasound treated waxy rice flour blends ".
We have responded to all the questions put forward by the reviewers with answers or explanations, and have revised our paper accordingly. The corrections were shown in red. We hope this version would meet with your approval. But, we are willing to do further revision if there is something deficiency.
We thank the reviewers for their valuable advice, and are grateful to you for all the favor you have done for us.
Sincerely yours,
Yuxing Wang
The following are our answers to the questions put forward by the reviewer:
Question 1: Rewrite the manuscript with professional English editing to enhance clarity and scientific rigor.
Answer 1: Thank you for pointing out the problems of the manuscript in terms of English usage, tense consistency and scientific expression. We have conducted a comprehensive language edit of the entire text, with a focus on correcting grammatical errors, unifying tenses, breaking down lengthy sentences, and improving the accuracy of professional terms to enhance the readability and scientific nature of the text.
Question 2: Strengthen the introduction and rationale, explaining the novelty and why ultrasound + HACS is investigated together.
Answer 2: Thank you for your suggestion. Firstly, in the introduction, we supplemented the advantages of physical ultrasonic modification over other chemical and biological modifications, and the specific modification is: "Most current research focuses on improving the functional properties of starch through physical, chemical or biological means. However, due to issues such as residual chemical reagents, complex processes and high costs, the application of chemical modification and enzymatic modification methods in the food processing field has been limited to a certain extent. In contrast, physical modification techniques do not require exogenous chemical substances, are easy to operate, and have high safety, which better meets the green development needs of the food industry. Among these, ultrasonic treatment has become a research focus in recent years for its high efficiency and controllability." (Page 2 Line 41-48). However, single ultrasound has limited effect on reducing starch digestibility, so new methods need to be explored.
Then, by comparing HACS with common starch, we further highlighted the advantages of HACS in terms of heat resistance, digestion resistance and practical application. The revised content is:"The amylose content in high amylose corn starch (HACS) is greater than 70%. Compared with common starch, HACS exhibits stronger heat resistance. After high-temperature treatment, more RS components are retained, thereby conferring higher anti-digestive properties and a lower GI. In recent years, HACS has been widely used in food preparation due to these advantages" (Page 2 Line 64-68). This makes our reasons for choosing HACS more solid and the logic clearer. Next, we further proposed the hypothesis of the synergistic effect of two physical methods, namely ultrasonic action and the combined addition of HACS, and studied the variation patterns of their influences on the physicochemical properties of glutinous rice flour and the digestion characteristics of rice cakes. This is done to better demonstrate the novelty and practical value of the work in this paper, making the logic of the research background and innovation points clearer.
Question 3: Include or clarify the experimental methodology in detail, especially regarding ultrasound treatment and digestibility assays.
Answer 3: Thank you for your suggestion. We further revised and supplemented some parts of the experimental method, including clarifying the frequency of ultrasonic treatment (25 kHz), the detailed steps for amylose determination (Page 4 Line 111-120), the heating and cooling parameters of the RVA program (Page 6 Line 147-150), the instrument specifications and scanning range for rheological testing (Page 6 Line 153-161), and the experimental steps and calculation methods of the DNS method used in in vitro digestion (Page 8 Line 202-208). And some other experimental details. In addition, we have also supplemented the data processing methods (Page 9 Line 227). All corrected parts are indicated in red. We believe that the modifications to these parts have enhanced the reproducibility of the experimental methods and the rigor of the paper.
Question 4: Clearly distinguish effects due to ultrasound vs. HACS vs. their interaction.
Answer 4: Thank you for your valuable suggestions. This train of thought is highly inspiring for enhancing the rigor of research.It should be noted that ultrasonic treatment of glutinous rice flour is a research work that our research group has completed in the early stage (Reference 8). The core research content of this paper is to further incorporate HACS on the basis of the previous ultrasonic treatment, with a focus on exploring the physicochemical properties and digestive characteristics changes of the HACS-UWRF blend system. Therefore, in the experimental design of this study, the influence of HACS acting alone was ignored. At present, due to the limitations of sample conditions and time, it is temporarily impossible to supplement relevant control experiments. However, in the discussion section, we will conduct an in-depth analysis of the possible mechanisms of action of both in combination with existing literature. The viewpoint you put forward is very important. This viewpoint will serve as an important basis for our future research design. Thank you again for your meticulous guidance.
Question 5: Provide comprehensive statistical analysis, include error bars, and define significance.
Answer 5: Thank you for your suggestion. We further modified it in 2.13.Statistical analysis. The revised content is: "statistical analysis was performed using IBM SPSS 19.0, including one-way analysis of variance (ANOVA) and Duncan’s multiple range test. Differences were considered significant at P < 0.05, and results were expressed as mean±standard deviation". Meanwhile, we have rechecked the significance levels of each indicator in the Results and discussion section and the error line markings in the charts to confirm that all contents are accurate.
Question 6: Improve mechanistic discussion using supportive literature and techniques (e.g., FTIR, XRD, microscopy).
Answer 6: Thank you for your suggestions. We have made targeted revisions based on this advice: On the one hand, we have further analyzed the experimental results of each part in depth, supplemented multiple relevant literatures as theoretical support, and systematically explained the potential mechanisms behind the experimental phenomena; On the other hand, we have closely linked the test results of amylose content, WSI, SP, gelatinization characteristics, thermal performance, structural performance (XRD and FTIR), SEM morphology and in vitro digestibility, etc., to clearly present the intrinsic correlation among different properties. All corrections have been marked in red. We believe these adjustments have made the logical chain of each part more coherent and the mechanistic discussions more thorough.
Question 7: Temper tbhe conclusions — suggest potential applications without overpromising or extrapolating eyond data.
Answer 7: Thank you for your suggestion. We have deleted the inappropriate expressions in the article. In addition, based on the results of the previous integrated analysis, we have also made corresponding modifications to the conclusion section. The specific modifications are as follows: "The addition of HACS significantly affected the physicochemical properties and in vitro digestibility of the HACS-UWRF blend system. With increasing HACS content, amylose content significantly increased (P < 0.05), which compacted the internal structure of starch molecules, gradually increased the average particle size, and smoothed the surface of starch particles. Meanwhile, HACS alleviated the rupture of starch particles in the blends, inhibiting starch particle water absorption and expansion. This reduced the gelatinization viscosity of the mixed powder, ultimately enhancing the thermal paste stability and shear resistance of the starch and suppressing starch retrogradation and regeneration. XRD and FTIR results further indicated that all blends exhibited typical A-type crystallization. HACS addition enhanced the RC and short-range ordered structure of starch. Rheological results showed that the starch paste of the blends is a pseudoplastic non-Newtonian fluid, exhibiting shear-thinning behavior with the system being mainly elastic. In vitro digestibility results revealed that with increasing HACS content, the RS content of rice cakes significantly increased from 53.57%±0.65 to 63.83%±0.53 (P < 0.05), and the eGI value significantly decreased from 69.16±0.15 to 58.97±0.01 (P < 0.05). Rice cakes belong to medium-GI foods. This research can be applied to develop glutinous rice starch foods with slow-glycemic increase properties."
